# Encoding High Dimensional Local Features by Sparse Coding Based Fisher Vectors

**Lingqiao Liu**[1], **Chunhua Shen**[1,2], **Lei Wang**[3], **Anton van den Hengel**[1,2], **Chao Wang**[3]

[1] School of Computer Science, University of Adelaide, Australia

[2] ARC Centre of Excellence for Robotic Vision

[3] School of Computer Science and Software Engineering, University of Wollongong, Australia

## Abstract

Deriving from the gradient vector of a generative model of local features, Fisher vector coding (FVC) has been identified as an effective coding method for image classification. Most, if not all, FVC implementations employ the Gaussian mixture model (GMM) to characterize the generation process of local features. This choice has shown to be sufficient for traditional low dimensional local features, e.g., SIFT; and typically, good performance can be achieved with only a few hundred Gaussian distributions. However, the same number of Gaussians is insufficient to model the feature space spanned by higher dimensional local features, which have become popular recently. In order to improve the modeling capacity for high dimensional features, it turns out to be inefficient and computationally impractical to simply increase the number of Gaussians.

In this paper, we propose a model in which each local feature is drawn from a Gaussian distribution whose mean vector is sampled from a subspace. With certain approximation, this model can be converted to a sparse coding procedure and the learning/inference problems can be readily solved by standard sparse coding methods. By calculating the gradient vector of the proposed model, we derive a new fisher vector encoding strategy, termed Sparse Coding based Fisher Vector Coding (SCFVC). Moreover, we adopt the recently developed Deep Convolutional Neural Network (CNN) descriptor as a high dimensional local feature and implement image classification with the proposed SCFVC. Our experimental evaluations demonstrate that our method not only significantly outperforms the traditional GMM based Fisher vector encoding but also achieves the state-of-the-art performance in generic object recognition, indoor scene, and fine-grained image classification problems.

## 1 Introduction

Fisher vector coding is a coding method derived from the Fisher kernel [1] which was originally proposed to compare two samples induced by a generative model. Since its introduction to computer vision [2], many improvements and variants have been proposed. For example, in [3] the normalization of Fisher vectors is identified as an essential step to achieve good performance; in [4] the spatial information of local features is incorporated; in [5] the model parameters are learned through a end-to-end supervised training algorithm and in [6] multiple layers of Fisher vector coding modules are stacked into a deep architecture. With these extensions, Fisher vector coding has been established as the state-of-the-art image classification approach.

Almost all of these methods share one common component: they all employ Gaussian mixture model (GMM) as the generative model for local features. This choice has been proved effective in modeling standard local features such as SIFT, which are often of low dimension. Usually, using a

mixture of a few hundred Gaussians has been sufficient to guarantee good performance. Generally speaking, the distribution of local features can only be well captured by a Gaussian distribution within a local region due to the variety of local feature appearances and thus the number of Gaussian mixtures needed is essentially determined by the volume of the feature space of local features.

Recently, the choice of local features has gone beyond the traditional local patch descriptors such as SIFT or SURF [7] and higher dimensional local features such as the activation of a pre-trained deep neural-network [8] or pooled coding vectors from a local region [9, 10] have demonstrated promising performance. The higher dimensionality and rich visual content captured by those features make the volume of their feature space much larger than that of traditional local features. Consequently, a much larger number of Gaussian mixtures will be needed in order to model the feature space accurately. However, this would lead to the explosion of the resulted image representation dimensionality and thus is usually computationally impractical.

To alleviate this difficulty, here we propose an alternative solution. We model the generation process of local features as randomly drawing features from a Gaussian distribution whose mean vector is randomly drawn from a subspace. With certain approximation, we convert this model to a sparse coding model and leverage an off-the-shelf solver to solve the learning and inference problems. With further derivation, this model leads to a new Fisher vector coding algorithm called Sparse Coding based Fisher Vector Coding (SCFVC). Moreover, we adopt the recently developed Deep Convolutional Neural Network to generate regional local features and apply the proposed SCFVC to these local features to build an image classification system.

To demonstrate its effectiveness in encoding the high dimensional local feature, we conduct a series of experiments on generic object, indoor scene and fine-grained image classification datasets, it is shown that our method not only significantly outperforms the traditional GMM based Fisher vector coding in encoding high dimensional local features but also achieves state-of-the-art performance in these image classification problems.

## 2   Fisher vector coding

### 2.1   General formulation

Given two samples generated from a generative model, their similarity can be evaluated by using a Fisher kernel [1]. The sample can take any form, including a vector or a vector set, as long as its generation process can be modeled. For a Fisher vector based image classification approach, the sample is a set of local features extracted from an image which we denote it as $\mathbf{X} = \{\mathbf{x}_1, \mathbf{x}_2, \cdots, \mathbf{x}_T\}$. Assuming each $\mathbf{x}_i$ is modeled by a p.d.f $P(\mathbf{x}|\lambda)$ and is drawn i.i.d, in Fisher kernel a sample $\mathbf{X}$ can be described by the gradient vector over the model parameter $\lambda$

$$\mathbf{G}_\lambda^\mathbf{X} = \nabla_\lambda \log P(\mathbf{X}|\lambda) = \sum_i \nabla_\lambda \log P(\mathbf{x}_i|\lambda). \tag{1}$$

The Fisher kernel is then defined as $K(\mathbf{X}, \mathbf{Y}) = \mathbf{G}_\lambda^{\mathbf{X}^T} \mathbf{F}^{-1} \mathbf{G}_\lambda^\mathbf{X}$, where $\mathbf{F}$ is the information matrix and is defined as $\mathbf{F} = E[\mathbf{G}_\lambda^\mathbf{X} \mathbf{G}_\lambda^{\mathbf{X}^T}]$. In practice, the role of the information matrix is less significant and is often omitted for computational simplicity [3]. As a result, two samples can be directly compared by the linear kernel of their corresponding gradient vectors which are often called Fisher vectors. From a bag-of-features model perspective, the evaluation of the Fisher kernel for two images can be seen as first calculating the gradient or Fisher vector of each local feature and then performing sum-pooling. In this sense, the Fisher vector calculation for each local feature can be seen as a coding method and we call it Fisher vector coding in this paper.

### 2.2   GMM based Fisher vector coding and its limitation

To implement the Fisher vector coding framework introduced above, one needs to specify the distribution $P(\mathbf{x}|\lambda)$. In the literature, most, if not all, works choose GMM to model the generation process of $\mathbf{x}$, which can be described as follows:

- Draw a Gaussian model $\mathcal{N}(\mu_k, \Sigma_k)$ from the prior distribution $P(k), \ k = 1, 2, \cdots, m$ .
- Draw a local feature $\mathbf{x}$ from $\mathcal{N}(\mu_k, \Sigma_k)$.

Generally speaking, the distribution of $\mathbf{x}$ resembles a Gaussian distribution only within a local region of feature space. Thus, for a GMM, each of Gaussian essentially models a small partition of the feature space and many of them are needed to depict the whole feature space. Consequently, the number of mixtures needed will be determined by the volume of the feature space. For the commonly used low dimensional local features, such as SIFT, it has been shown that it is sufficient to set the number of mixtures to few hundreds. However, for higher dimensional local features this number may be insufficient. This is because the volume of feature space usually increases quickly with the feature dimensionality. Consequently, the same number of mixtures will result in a coarser partition resolution and imprecise modeling.

To increase the partition resolution for higher dimensional feature space, one straightforward solution is to increase the number of Gaussians. However, it turns out that the partition resolution increases slowly (compared to our method which will be introduced in the next section) with the number of mixtures. In other words, much larger number of Gaussians will be needed and this will result in a Fisher vector whose dimensionality is too high to be handled in practice.

## 3 Our method

### 3.1 Infinite number of Gaussians mixture

Our solution to this issue is to go beyond a fixed number of Gaussian distributions and use an infinite number of them. More specifically, we assume that a local feature is drawn from a Gaussian distribution with a randomly generated mean vector. The mean vector is a point on a subspace spanned by a set of bases (which can be complete or over-complete) and is indexed by a latent coding vector $\mathbf{u}$. The detailed generation process is as follows:

- Draw a coding vector $\mathbf{u}$ from a zero mean Laplacian distribution $P(\mathbf{u}) = \frac{1}{2\lambda} \exp(-\frac{|\mathbf{u}|}{\lambda})$.
- Draw a local feature $\mathbf{x}$ from the Gaussian distribution $\mathcal{N}(\mathbf{Bu}, \Sigma)$,

where the Laplace prior for $P(\mathbf{u})$ ensures the sparsity of resulting Fisher vector which can be helpful for coding. Essentially, the above process resembles a sparse coding model. To show this relationship, let's first write the marginal distribution of $\mathbf{x}$:

$$P(\mathbf{x}) = \int_{\mathbf{u}} P(\mathbf{x}, \mathbf{u}|\mathbf{B})d\mathbf{u} = \int_{\mathbf{u}} P(\mathbf{x}|\mathbf{u}, \mathbf{B})P(\mathbf{u})d\mathbf{u}. \tag{2}$$

The above formulation involves an integral operator which makes the likelihood evaluation difficult. To simplify the calculation, we use the point-wise maximum within the integral term to approximate the likelihood, that is,

$$P(\mathbf{x}) \approx P(\mathbf{x}|\mathbf{u}^*, \mathbf{B})P(\mathbf{u}^*).$$
$$\mathbf{u}^* = \underset{\mathbf{u}}{\operatorname{argmax}} P(\mathbf{x}|\mathbf{u}, \mathbf{B})P(\mathbf{u}) \tag{3}$$

By assuming that $\Sigma = diag(\sigma_1^2, \cdots, \sigma_m^2)$ and setting $\sigma_1^2 = \cdots = \sigma_m^2 = \sigma^2$ as a constant. The logarithm of $P(\mathbf{x})$ is written as

$$\log(P(\mathbf{x}|\mathbf{B})) = \min_{\mathbf{u}} \frac{1}{\sigma^2} \|\mathbf{x} - \mathbf{Bu}\|_2^2 + \lambda \|\mathbf{u}\|_1, \tag{4}$$

which is exactly the objective value of a sparse coding problem. This relationship suggests that we can learn the model parameter $\mathbf{B}$ and infer the latent variable $\mathbf{u}$ by using the off-the-shelf sparse coding solvers.

One question for the above method is that compared to simply increasing the number of models in traditional GMM, how much improvement is achieved by increasing the partition resolution. To answer this question, we designed an experiment to compare these two schemes. In our experiment, the partition resolution is roughly measured by the average distance (denoted as $d$) between a feature and its closest mean vector in the GMM or the above model. The larger $d$ is, the lower the partition resolution is. The comparison is shown in Figure 1. In Figure 1 (a), we increase the dimensionality

of local features [1] and for each dimension we calculate $d$ in a GMM model with 100 mixtures. As seen, $d$ increases quickly with the feature dimensionality. In Figure 1 (b), we try to reduce $d$ by introducing more mixture distributions in GMM model. However, as can be seen, $d$ drops slowly with the increase in the number of Gaussians. In contrast, with the proposed method, we can achieve much lower $d$ by using only 100 bases. This result illustrates the advantage of our method.

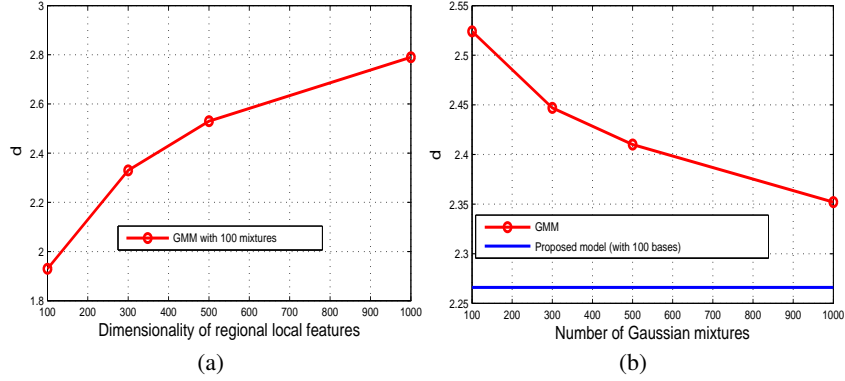

Figure 1: Comparison of two ways to increase the partition resolution. (a) For GMM, $d$ (the average distance between a local feature and its closest mean vector) increases with the local feature dimensionality. Here the GMM is fixed at 100 Gaussians. (b) $d$ is reduced in two ways (1) simply increasing the number of Gaussian distributions in the mixture. (2) using the proposed generation process. As can be seen, the latter achieves much lower $d$ even with a small number of bases.

### 3.2 Sparse coding based Fisher vector coding

Once the generative model of local features is established, we can readily derive the corresponding Fisher coding vector by differentiating its log likelihood, that is,

$$\mathcal{C}(\mathbf{x}) = \frac{\partial \log(P(\mathbf{x}|\mathbf{B}))}{\partial \mathbf{B}} = \frac{\partial \frac{1}{\sigma^2} \|\mathbf{x} - \mathbf{B}\mathbf{u}^*\|_2^2 + \lambda \|\mathbf{u}^*\|_1}{\partial \mathbf{B}}$$
$$\mathbf{u}^* = \operatorname*{argmax}_{\mathbf{u}} P(\mathbf{x}|\mathbf{u}, \mathbf{B}) P(\mathbf{u}). \tag{5}$$

Note that the differentiation involves $\mathbf{u}^*$ which implicitly interacts with $\mathbf{B}$. To calculate this term, we notice that the sparse coding problem can be reformulated as a general quadratic programming problem by defining $\mathbf{u}^+$ and $\mathbf{u}^-$ as the positive and negative parts of $\mathbf{u}$, that is, the sparse coding problem can be rewritten as

$$\min_{\mathbf{u}^+, \mathbf{u}^-} \|\mathbf{x} - \mathbf{B}(\mathbf{u}^+ - \mathbf{u}^-)\|_2^2 + \lambda \mathbf{1}^T (\mathbf{u}^+ + \mathbf{u}^-)$$
$$s.t.\ \mathbf{u}^+ \geq 0 \quad \mathbf{u}^- \geq 0 \tag{6}$$

By further defining $\mathbf{u}' = (\mathbf{u}^+, \mathbf{u}^-)^T$, $\log(P(\mathbf{x}|\mathbf{B}))$ can be expressed in the following general form,

$$\log(P(\mathbf{x}|\mathbf{B})) = \mathcal{L}(\mathbf{B}) = \max_{\mathbf{u}'}\ \mathbf{u}'^T \mathbf{v}(\mathbf{B}) - \frac{1}{2}\mathbf{u}'^T \mathbf{P}(\mathbf{B})\mathbf{u}', \tag{7}$$

where $\mathbf{P}(\mathbf{B})$ and $\mathbf{v}(\mathbf{B})$ are a matrix term and a vector term depending on $\mathbf{B}$ respectively. The derivative of $\mathcal{L}(\mathbf{B})$ has been studied in [11]. According to the Lemma 2 in [11], we can differentiate $\mathcal{L}(\mathbf{B})$ with respect to $\mathbf{B}$ as if $\mathbf{u}'$ did not depend on $\mathbf{B}$. In other words, we can firstly calculate $\mathbf{u}'$ or equivalently $\mathbf{u}^*$ by solving the sparse coding problem and then obtain the Fisher vector $\frac{\partial \log(P(\mathbf{x}|\mathbf{B}))}{\partial \mathbf{B}}$ as

$$\frac{\partial \frac{1}{\sigma^2} \|\mathbf{x} - \mathbf{B}\mathbf{u}^*\|_2^2 + \lambda \|\mathbf{u}^*\|_1}{\partial \mathbf{B}} = (\mathbf{x} - \mathbf{B}\mathbf{u}^*)\mathbf{u}^{*T}. \tag{8}$$

Table 1: Comparison of results on Pascal VOC 2007. The lower part of this table lists some results reported in the literature. We only report the mean average precision over 20 classes. The average precision for each class is listed in Table 2.

| Methods | mean average precision | Comments |
|---|---|---|
| SCFVC (proposed) | **76.9%** | single scale, no augmented data |
| GMMFVC | 73.8% | single scale, no augmented data |
| CNNaug-SVM [8] | **77.2%** | with augmented data, use CNN for whole image |
| CNN-SVM [8] | 73.9% | no augmented data.use CNN for whole image |
| NUS [13] | 70.5% | - |
| GHM [14] | 64.7% | - |
| AGS [15] | 71.1% | - |

Note that the Fisher vector expressed in Eq. (8) has an interesting form: it is simply the outer product between the sparse coding vector $\mathbf{u}^*$ and the reconstruction residual term $(\mathbf{x} - \mathbf{B}\mathbf{u}^*)$. In traditional sparse coding, only the $k$th dimension of a coding vector $u_k$ is used to indicate the relationship between a local feature $\mathbf{x}$ and the $k$th basis. Here in the sparse coding based Fisher vector, the coding value $u_k$ multiplied by the reconstruction residual is used to capture their relationship.

### 3.3 Pooling and normalization

From the i.i.d assumption in Eq. (1), the Fisher vector of the whole image is [2]

$$\frac{\partial \log(P(\mathcal{I}|\mathbf{B}))}{\partial \mathbf{B}} = \sum_{\mathbf{x}_i \in \mathcal{I}} \frac{\partial \log(P(\mathbf{x}_i|\mathbf{B}))}{\partial \mathbf{B}} = \sum_{\mathbf{x}_i \in \mathcal{I}} (\mathbf{x}_i - \mathbf{B}\mathbf{u}_i^*)\mathbf{u}_i^{*\top}. \tag{9}$$

This is equivalent to performing the sum-pooling for the extracted Fisher coding vectors. However, it has been observed [3, 12] that the image signature obtained using sum-pooling tends to over-emphasize the information from the background [3] or bursting visual words [12]. It is important to apply normalization when sum-pooling is used. In this paper, we apply intra-normalization [12] to normalize the pooled Fisher vectors. More specifically, we apply $l2$ normalization to the subvectors $\sum_{\mathbf{x}_i \in \mathcal{I}} (\mathbf{x}_i - \mathbf{B}\mathbf{u}_i^*)u_{i,k}^* \ \forall k$, where $k$ indicates the $k$th dimension of the sparse coding $\mathbf{u}_i^*$. Besides intra-normalization, we also utilize the power normalization as suggested in [3].

### 3.4 Deep CNN based regional local features

Recently, the middle-layer activation of a pre-trained deep CNN has been demonstrated to be a powerful image descriptor [8, 16]. In this paper, we employ this descriptor to generate a number of local features for an image. More specifically, an input image is first resized to $512 \times 512$ pixels and regions with the size of $227 \times 227$ pixels are cropped with the stride 8 pixels. These regions are subsequently feed into the deep CNN and the activation of the sixth layer is extracted as local features for these regions. In our implementation, we use the Caffe [17] package which provides a deep CNN pre-trained on ILSVRC2012 dataset and its 6-th layer is a 4096-dimensional vector. This strategy has demonstrated better performance than directly using deep CNN features for the whole image recently [16].

Once regional local features are extracted, we encoded them using the proposed SCFVC method and generate an image level representation by sum-pooling and normalization. Certainly, our method is open to the choice of other high-dimensional local features. The reason for choosing deep CNN features in this paper is that by doing so we can demonstrate state-of-the-art image classification performance.

## 4 Experimental results

We conduct experimental evaluation of the proposed sparse coding based Fisher vector coding (SCFVC) on three large datasets: Pascal VOC 2007, MIT indoor scene-67 and Caltech-UCSD Birds-

Table 2: Comparison of results on Pascal VOC 2007 for each of 20 classes. Besides the proposed SCFVC and the GMMFVC baseline, the performance obtained by directly using CNN as global feature is also compared.

|  | aero | bike | bird | boat | bottle | bus | car | cat | chair | cow |
|---|---|---|---|---|---|---|---|---|---|---|
| SCFVC | **89.5** | **84.1** | **83.7** | **83.7** | **43.9** | **76.7** | **87.8** | **82.5** | **60.6** | **69.6** |
| GMMFVC | 87.1 | 80.6 | 80.3 | 79.7 | 42.8 | 72.2 | 87.4 | 76.1 | 58.6 | 64.0 |
| CNN-SVM | 88.5 | 81.0 | 83.5 | 82.0 | 42.0 | 72.5 | 85.3 | 81.6 | 59.9 | 58.5 |
|  | table | dog | horse | mbike | person | plant | sheep | sofa | train | TV |
| SCFVC | **72.0** | **77.1** | **88.7** | **82.1** | **94.4** | **56.8** | **71.4** | **67.7** | **90.9** | **75.0** |
| GMMFVC | 66.9 | 75.1 | 84.9 | 81.2 | 93.1 | 53.1 | 70.8 | 66.2 | 87.9 | 71.3 |
| CNN-SVM | 66.5 | 77.8 | 81.8 | 78.8 | 90.2 | 54.8 | 71.1 | 62.6 | 87.2 | 71.8 |

Table 3: Comparison of results on MIT-67. The lower part of this table lists some results reported in the literature.

| Methods | Classification Accuracy | Comments |
|---|---|---|
| SCFVC (proposed) | **68.2%** | with single scale |
| GMMFVC | 64.3% | with single scale |
| MOP-CNN [16] | **68.9%** | with three scales |
| VLAD level2 [16] | 65.5% | with single best scale |
| CNN-SVM [8] | 58.4% | use CNN for whole image |
| FV+Bag of parts [19] | 63.2% | - |
| DPM [20] | 37.6% | - |

200-2011. These are commonly used evaluation benchmarks for generic object classification, scene classification and fine-grained image classification respectively. The focus of these experiments is to examine that whether the proposed SCFVC outperforms the traditional Fisher vector coding in encoding high dimensional local features.

## 4.1 Experiment setup

In our experiments, the activations of the sixth layer of a pre-trained deep CNN are used as regional local features. PCA is applied to further reduce the regional local features from 4096 dimensions to 2000 dimensions. The number of Gaussian distributions and the codebook size for sparse coding is set to 100 throughout our experiments unless otherwise mentioned.

For the sparse coding, we use the algorithm in [18] to learn the codebook and perform the coding vector inference. For all experiments, linear SVM is used as the classifier.

## 4.2 Main results

**Pascal-07** Pascal VOC 2007 contains 9963 images with 20 object categories which form 20 binary (object vs. non-object) classification tasks. The use of deep CNN features has demonstrated the state-of-the-art performance [8] on this dataset. In contrast to [8], here we use the deep CNN features as *local features* to model a set of image regions rather than as a global feature to model the whole image. The results of the proposed SCFVC and traditional Fisher vector coding, denoted as GMMFVC, are shown in Table 1 and Table 2. As can be seen from Table 1, the proposed SCFVC leads to superior performance over the traditional GMMFVC and outperforms GMMFVC by 3%. By cross-referencing Table 2, it is clear that the proposed SCFVC outperforms GMMFVC in all of 20 categories. Also, we notice that the GMMFVC is merely comparable to the performance of directly using deep CNN as global features, namely, CNN-SVM in Table 1. Since both the proposed SCFVC and GMMFVC adopt deep CNN features as local features, this observation suggests that the advantage of using deep CNN features as local features can only be clearly demonstrated when the appropriate coding method, i.e. the proposed SCFVC is employed. Note that to further boost the

Table 4: Comparison of results on Birds-200 2011. The lower part of this table lists some results reported in the literature.

| Methods | Classification Accuracy | Comments |
| --- | --- | --- |
| SCFVC (proposed) | **66.4%** | with single scale |
| GMMFVC | 61.7% | with single scale |
| CNNaug-SVM [8] | 61.8% | with augmented data, use CNN for the whole image |
| CNN-SVM [8] | 53.3% | no augmented data, use CNN as global features |
| DPD+CNN+LogReg [21] | 65.0% | use part information |
| DPD [22] | 51.0% | - |

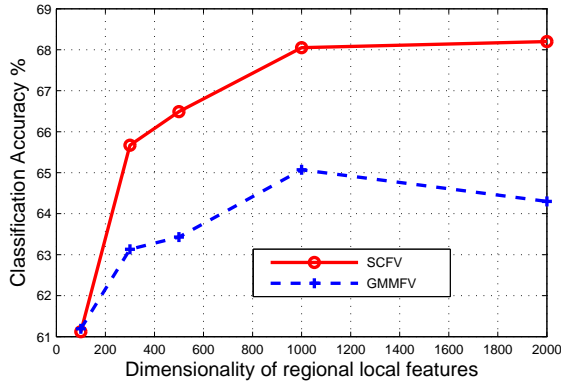

Figure 2: The performance comparison of classification accuracy vs. local feature dimensionality for the proposed SCFVC and GMMFVC on MIT-67.

performance, one can adopt some additional approaches like introducing augmented data or combining multiple scales. Some of the methods compared in Table 1 have employed these approaches and we have commented this fact as so inform readers that whether these methods are directly comparable to the proposed SCFVC. We do not pursue these approaches in this paper since the focus of our experiment is to compare the proposed SCFVC against traditional GMMFVC.

**MIT-67** MIT-67 contains 6700 images with 67 indoor scene categories. This dataset is quite challenging because the differences between some categories are very subtle. The comparison of classification results are shown in Table 3. Again, we observe that the proposed SCFVC significantly outperforms traditional GMMFVC. To the best of our knowledge, the best performance on this dataset is achieved in [16] by concatenating the features extracted from three different scales. The proposed method achieves the same performance only using a single scale. We also tried to concatenate the image representation generated from the proposed SCFVC with the global deep CNN feature. The resulted performance can be as high as **70%** which is by far the best performance achieved on this dataset.

**Birds-200-2011** Birds-200-2011 contains 11788 with 200 different birds species, which is a commonly used benchmark for fine-grained image classification. The experimental results on this dataset are shown in Table 4. The advantage of SCFVC over GMMFVC is more pronounced on this dataset: SCFVC outperforms GMMFVC by over 4%. We also notice two interesting observations: (1) GMMFVC even achieves comparable performance to the method of using the global deep CNN feature with augmented data, namely, CNNaug-SVM in Table 4. (2) Although we do not use any parts information (of birds), our method outperforms the result using parts information (DPD+CNN+LogReg in Table 4). These two observations suggest that using deep CNN features as local features is better for fine-grained problems and the proposed method can further boost its advantage.

Table 5: Comparison of results on MIT-67 with three different settings: (1) 100-basis codebook with 1000 dimensional local features, denoted as SCFV-100-1000D (2) 400 Gaussian mixtures with 300 dimensional local features, denoted as GMMFV-400-300D (3) 1000 Gaussian mixtures with 100 dimensional local features denoted as GMMFV-1000-100D. They have the same/similar total image representation dimensionality.

| SCFV-100-1000D | GMMFV-400-300D | GMMFV-1000-100D |
|---|---|---|
| **68.1%** | 64.0% | 60.8% |

### 4.3 Discussion

In the above experiments, the dimensionality of local features is fixed to 2000. But how about the performance comparison between the proposed SCFV and traditional GMMFV on lower dimensional features? To investigate this issue, we vary the dimensionality of the deep CNN features from 100 to 2000 and compare the performance of the two Fisher vector coding methods on MIT-67. The results are shown in Figure 2. As can be seen, for lower dimensionality (like 100), the two methods achieve comparable performance and in general both methods benefit from using higher dimensional features. However, for traditional GMMFVC, the performance gain obtained from increasing feature dimensionality is lower than that obtained by the proposed SCFVC. For example, from 100 to 1000 dimensions, the traditional GMMFVC only obtains 4% performance improvement while our SCFVC achieves 7% performance gain. This validates our argument that the proposed SCFVC is especially suited for encoding high dimensional local features.

Since GMMFVC works well for lower dimensional features, how about reducing the higher dimensional local features to lower dimensions and use more Gaussian mixtures? Will it be able to achieve comparable performance to our SCFVC which uses higher dimensional local features but a smaller number of bases? To investigate this issue, we also evaluate the classification performance on MIT-67 using 400 Gaussian mixtures with 300-dimension local features and 1000 Gaussian mixtures with 100-dimension local features. Thus the total dimensionality of these two image representations will be similar to that of our SCFVC which uses 100 bases and 1000-dimension local features. The comparison is shown in Table 5. As can be seen, the performance of these two settings are much inferior to the proposed one. This suggests that some discriminative information may have already been lost after the PCA dimensionality reduction and the discriminative power can not be re-boosted by simply introducing more Gaussian distributions. This verifies the necessity of using high dimensional local features and justifies the value of the proposed method.

In general, the inference step in sparse coding can be slower than the membership assignment in GMM model. However, the computational efficiency can be greatly improved by using an approximated sparse coding algorithm such as learned FISTA [23] or orthogonal matching pursuit [10]. Also, the proposed method can be easily generalized to several similar coding models, such as local linear coding [24]. In that case, the computational efficiency is almost identical (or even faster if approximated k-nearest neighbor algorithms are used) to the traditional GMMFVC.

## 5 Conclusion

In this work, we study the use of Fisher vector coding to encode high-dimensional local features. Our main discovery is that traditional GMM based Fisher vector coding is not particular well suited to modeling high-dimensional local features. As an alternative, we proposed to use a generation process which allows the mean vector of a Gaussian distribution to be chosen from a point in a subspace. This model leads to a new Fisher vector coding method which is based on sparse coding model. Combining with the activation of the middle layer of a pre-trained CNN as high-dimensional local features, we build an image classification system and experimentally demonstrate that the proposed coding method is superior to the traditional GMM in encoding high-dimensional local features and can achieve state-of-the-art performance in three image classification problems.

**Acknowledgements** This work was in part supported by Australian Research Council grants FT120100969, LP120200485, and the Data to Decisions Cooperative Research Centre. Correspondence should be addressed to C. Shen (email: chhshen@gmail.com).

## Footnotes

[1]This is achieved by performing PCA on a 4096-dimensional CNN regional descriptor. For more details about the feature we used, please refer to Section 3.4

[2]the vectorized form of $\frac{\partial \log(P(\mathcal{I}|\mathbf{B}))}{\partial \mathbf{B}}$ is used as the image representation.

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
