[Reviews · NeurIPS 2014]

Submitted by Assigned_Reviewer_11

This paper proposes a new Fisher vector coding method to encode high-dimensional local features. The proposed method assumes a generative model in which a local feature is drawn from a Gaussian distribution with a randomly generated mean vector. By approximating the likelihood, the resulting objective function becomes that of a sparse coding problem and the model parameters can be learnt by using the standard sparse coding solvers. A new Fisher coding vector can be derived by the differentiating the log likelihood of the generative model. The resulting vector is termed Sparse Coding based Fisher Vector Coding (SCFVC). In the experiments, FSFVC is applied to the Deep CNN descriptors and the FSFVC is better than the standard GMM based Fisher vector coding.

This paper is well written. It's easy to understand the proposed method. The originality is the combination of the well-known coding methods: the sparse coding and the Fisher vector coding. However there are some unclear points:

1. This paper doesn't define the Fisher Information Matrix (FIM) for the proposed model. Probably without the FIM, the resulting vector cannot called the Fisher vector. The metric in the parameter space (FIM) should be derived from the view of Information Geometry.

2. This paper does not mention the concrete computational complexity and speed of the proposed coding method except l.413-l.415. Also it would be better to discuss the scalability of the proposed coding method.

3. In eq. (8), this is represented by the matrix form not by the vector. How is the final form obtained?

4. In Table 2, the scores of the CNN-SVM in the bottle and the chair class are higher than the SCFVC. What do the bold types mean? The scores of the CNN-SVM seem much higher than others.

5. In the experiments, this paper compares the SCFVC with the GMMFVC, but there are no clear explanation about parameters of the GMMFVC except Table 5.

6. In MIT-67 dataset, the combination of the SCFVC and the deep CNN is the best performance. However, for the fair comparison, the SCFVC should be applied the three scales data.
Summary: It’s interesting to combine the sparse coding and the FV coding.

Submitted by Assigned_Reviewer_17

This paper proposes a method for fisher vector coding (FVC) of high dimensional local features. Most existing approaches that use FVC assume that local features are drawn from a Gaussian Mixture Model (GMM) where the number of mixture components is specified a priori. This paper argues that while this has worked well for FVC of low-dimensional local features like SIFT, it does not scale to high dimensional features because the number of Gaussians required to model the distribution of local features becomes prohibitive. Instead of using a GMM with a finite number of mixture components, this paper proposes to approximate an infinite Gaussian mixture using a set of basis vectors. Under the assumption of iid feature dimensions, the basis vectors and feature encoding can be learned using standard sparse coding. The paper provides derivation of the corresponding fisher vector, which can be computed analytically.

The paper is well-written. There are minor typos and grammatical errors throughout the paper, but they do not significantly impact comprehensibility. The proposed approach is technically sound and appears to yield good results empirically. One limitation of the approach that the paper glosses over is the assumption of a diagonal covariance matrix with constant sigma in deriving the log-likelihood in eq. 4. This is an important theoretical limitation, as GMMs can model correlation between feature dimensions, while the sparse coding based approach proposed in this paper does not. The authors should mention this limitation, and explicitly state that using PCA to first transform the local features is a practical method to get around this problem.

Experiments demonstrate that the proposed approach outperforms the use of GMM-based FVC and achieves state-of-the-art performance on several image classification, scene classification, and fine-grained image classification datasets. The paper investigates the tradeoff between increasing the number of mixture components versus using higher dimensional local features for GMM-based FVC. However, it would be even more informative if the paper could also provide a similar investigation on the tradeoff between the number of basis vectors and the dimensionality of local features for the proposed sparse coding FVC.

Overall, the paper contains good algorithmic novelty and demonstrates convincing and extensive empirical results. Hence, I would recommend that it be accepted for publication if the authors could provide an analysis of the tradeoff between the number of basis vectors and the dimensionality of local features for the proposed sparse coding FVC.

Minor Questions:
- For Pascal VOC 2007 results, there is a huge performance difference between FVC-based methods and CNN-SVM for bottle (in CNN-SVM's favor) and bus (in FVC's favor). Have you looked into the cause of these differences?

Typos:
- line 151: By "assuming" ...
- line 240: ... coding value u_k "multiplied by" the reconstruction residue ...
Summary: The paper proposes to a novel method for fisher vector coding (FVC) of high dimensional local features using sparse coding. It demonstrates convincing and extensive empirical results.

Submitted by Assigned_Reviewer_38

This paper proposes a sparse coding based fisher vector which is effective for encoding high dimensional local features such as Deep Convolutional Neural Network descriptors.

Overall, this paper is very interesting and very useful when we use DCNN descriptors as local feature descriptors. The effectiveness is clearly shown by the experiments with three common datasets.

Currently, DCNN descriptors become so common because of the release of open source DNN softwares such as Caffe and Overfeat. As a next research step, we have to tackle how to boost performance with DCNN features. In this sense, this paper is very interesting. I enjoyed reading the paper !

Pros)
1) The author proposed a novel extension of FV which is based on sparse coding. Although it is not so effective for low dimensional local features such as SIFT, they discover that it is very useful for high dimensional features such as DCNN descriptors. In addition to proposal of SCFVC, using DCNN descriptors as high dimensional local features is a novel and interesting idea as well. This might be a new trend in the community of object recognition.

2) The paper is well written and easy to read except Section 3. The analysis on the limitation of conventional FV and comparison between FV and the proposed SCFVC are described well.

3) The experimental results on three common datasets including Pascal VOC 2007, MIT indoor-67, and Caltech-UCSD Birds200 strongly supports the effectiveness of the proposed method.

4) Comparison between the proposed SCFV and traditional GMMFV from lower dimensional features to higher dimensional features is well analyzed in Section 4.3. This helps clarify the difference of the characteristics of two methods.

Cons)
1) It is not easy to understand the section which explains the proposed method. I could not understand why Eq.(4) was derived from (3). How do you obtain a model parameter B ? (In fact, I'm not so familiar with sparse coding.)

Comments
1) How did you decide the values of two hyper-parameters, $\sigma$ and $\lambda$?
How do they affect the results ?
Summary: This paper proposes a novel sparse coding based fisher vector which is effective for encoding high dimensional local features. In addition, using DCNN descriptors as high dimensional local features is a novel and interesting idea.

Submitted by Meta_Reviewer_8

Additional comments by AC

- Having obtained the coded features u_i, it seems bizarre not to
include them directly in your output feature set, e.g. as sum_i v_i or
perhaps in some more normalized form such as counting "significantly
nonzero" dimensions. The signs of the residuals x-B*u will vary
randomly so as things stand you appear to be throwing away almost all
of the direct information about your input feature activations. This
is likely to harm the performance.

- Terminology. Just FYI, in statistics derivatives of log likelihoods
wrt parameters are usually just called "scores", while the qualifier
"Fisher" implies that their covariance (the Fisher information) has
been used to normalize or kernelize them to get more meaningful
"information geometric" quantities. So at a pedantic level
you are using "sparse coding likelihood scores" but they are not
"Fisher" ones. Similarly, accumulating scores over samples is called
"scoring" so the whole activity of the paper could be summarized as
"... by sparse coding likelihood scoring".
Summary: This is a decent poster on sparsifying likelihood-score based coding
for use with high-dimensional deep learning features. The results are
likely to improve if the input features are included in the final
classifier and this should be tested in the final paper.
Author Feedback
Author rebuttal: We thank all the valuable comments from all reviewers and address their concerns as follows:
For Assigned_Reviewer_11
1. About the information matrix
Yes, in the strict sense, the FIM needs to be defined, but in practice this matrix is often omitted and we also discussed this issue at lines 093-095. The reason of this treatment has been discussed in Ref [1] (below the equation (9)) and [3]. In this paper, we follow this convention for computational simplicity.

2.On the computational complexity and speed of the proposed coding method
Our method needs around 0.7s to generate the image representation from 200 regional-CNN features (un-optimized Matlab code). This speed can be greatly improved by using some accelerated or approximated versions of sparse coding as also discussed at lines 413-415.

3.The matrix form in Eq. (8)
The image representation is obtained by flattening the matrix form in Eq. (8). We will make this clear in our revised version.

4.The score in Table 2
Thanks for pointing out of this, which is a typo. The correct scores should be: ``bird’’ 83.5, ``boat’’ 82.0 ``bottle’’ 42.0 ``bus’’ 72.5 ``car’’ 85.3 ``cat’’ 81.6 ``chair’’ 59.9 ``cow’’ 58.5 ``table’’ 66.5 ``dog’’ 77.8 and ``horse’’ 81.8. We will correct this error in our revised version.

5.On the parameters of GMMFVC
The parameters of GMMFVC have been discussed at lines 307-313. We will rewrite this sentence to make it clearer.

6.On the performance on MIT-67
The 70% accuracy is obtained by combining two scales. Since the focus of this paper is to compare two different Fisher encoding methods (SCFV vs. GMMFVC) rather than producing best performance on particular datasets, we didn’t use three scales as in [16]. However, with two scales, out performance has been already better than that reported in Ref[16] (3 scales) and usually combining features form more scales only increases the performance.

For Assigned_Reviewer_17
7.“It would be even more informative if the paper could also provide a similar investigation the trade-off between the number of basis vectors and the dimensionality …”
Thanks for this suggestion. We will add this experiment in our revised version.

8.The performance difference in Table 2
Please see Q4.

For Assigned_Reviewer_38
9.“It is not easy to understand the section which explains ….”
Eq. (4) is derived by substituting the P.D.Fs of P(x|u,B) and P(u) to Eq.(3) (Note that we have defined P(x|u,B) as Gaussian distribution and P(u) as Laplacian distribution at lines 135-137 ). We will make this derivation clearer in our revised version. The codebook B is learned from a number of (40K) sampled local features by using the algorithm in [18] as mentioned on line 158 and line 312.

10.The setting of $\sigma$ and $\lambda$
These two hyper parameters can be merged into one, say, let $\hat{\sigma} = \sigma \lambda $ and its value can be determined by cross-validation. We use the same value for this hyper-parameter across three datasets presented in this paper.